# Spreading of P301S Aggregated Tau Investigated in Organotypic Mouse Brain Slice Cultures

**DOI:** 10.3390/biom12091164

**Published:** 2022-08-23

**Authors:** Dhwani S. Korde, Christian Humpel

**Affiliations:** Laboratory of Psychiatry and Experimental Alzheimer’s Research, Medical University of Innsbruck, 6020 Innsbruck, Austria

**Keywords:** Alzheimer, tauopathy, tau, organotypic brain slices, spreading, collagen hydrogels

## Abstract

Tau pathology extends throughout the brain in a prion-like fashion through connected brain regions. However, the details of the underlying mechanisms are incompletely understood. The present study aims to examine the spreading of P301S aggregated tau, a mutation that is implicated in tauopathies, using organotypic slice cultures. Coronal hippocampal organotypic brain slices (170 µm) were prepared from postnatal (day 8–10) C57BL6 wild-type mice. Collagen hydrogels loaded with P301S aggregated tau were applied to slices and the spread of tau was assessed by immunohistochemistry after 8 weeks in culture. Collagen hydrogels prove to be an effective protein delivery system subject to natural degradation in 14 days and they release tau proteins up to 8 weeks. Slices with un- and hyperphosphorylated P301S aggregated tau demonstrate significant spreading to the ventral parts of the hippocampal slices compared to empty collagen hydrogels after 8 weeks. Moreover, the spread of P301S aggregated tau occurs in a time-dependent manner, which was interrupted when the neuroanatomical pathways are lesioned. We illustrate that the spreading of tau can be investigated in organotypic slice cultures using collagen hydrogels to achieve a localized application and slow release of tau proteins. P301S aggregated tau significantly spreads to the ventral areas of the slices, suggesting that the disease-relevant aggregated tau form possesses spreading potential. Thus, the results offer a novel experimental approach to investigate tau pathology.

## 1. Introduction

Over 55 million people currently live with dementia worldwide and this number is only growing at a staggering rate that is predicted to reach 78 million and 135 million people by 2030 and 2050, respectively [1,2]. Alzheimer’s disease (AD) is the leading cause of dementia, contributing to 60–70% of cases and is the most common neurodegenerative disease. With an ageing human population, the healthcare, social and economic burden of AD will worsen in the upcoming years, underlining the imperative nature of research and development of therapeutic approaches.

AD is characterized by the accumulation of extracellular aggregates composed of amyloid-β (Aβ) protein, intracellular neurofibrillary tangles (NFTs) composed of hyperphosphorylated tau protein, neuropil threads and dystrophic neurites [3]. Additionally, astrogliosis, activation of microglia and atrophy owing to neuronal and synaptic loss are accompanying pathologies [4]. There is also evidence of a shared Aβ deposition underlying the pathogenesis of AD and cerebral amyloid angiopathy (CAA), highlighting the potential of cross-talk between neurodegenerative and cerebrovascular diseases [5].

### 1.1. Tau Protein and AD

Tau is a microtubule-associated protein that is predominantly found in the cytosol and axons of neurons. It plays a crucial role in regulating the dynamics of microtubules and thus maintaining the cytoskeletal integrity of the cell. It is encoded by the *MAPT* gene and is located on the chromosome 17 in humans. The gene spans across 16 exons that result in six different alternative splicing isoforms consisting of 351–441 amino acids [6]. Tau is an example of an intrinsically disordered protein with a greater predisposition to self-aggregate and form β-sheet structures, which are the principal constituents of NFTs [7].

Tau can contain either three or four microtubule-binding domain repeats (3R/4R), which serve the dual purpose of binding to microtubules and contributing to the propensity to aggregate into paired-helical filaments (PHFs), which are the building blocks of NFTs. The intrinsically disordered nature of tau makes it particularly amenable to various post-translational modifications, such as phosphorylation, acetylation, ubiquitination and proteolytic cleavage. Phosphorylation is the best-characterized post-translational modification. Hyperphosphorylation of tau is linked to a pathological state that potentially reduces the binding ability to microtubules and hence may affect the overall stability of microtubules [8]. Conversely, tau has also been implicated in playing a role in promoting the assembly of microtubules as opposed to its classical microtubule stabilizer role, highlighting that the underlying mechanism by which tau exerts its pathological effects and physiological functions of tau are yet to be fully uncovered [9]. Tau pathology is a hallmark of a group of neurodegenerative diseases known as tauopathies, including AD, frontotemporal dementia, progressive supranuclear palsy (PSP) and Pick’s disease, amongst others [10].

### 1.2. Tau and Propagation

Post-mortem studies have revealed that tau pathology in AD propagates throughout the brain in a stereotypical spatio-temporal pattern starting from the locus coeruleus to the entorhinal and transentorhinal cortex to hippocampal regions and finally moving on to the neocortical areas. The stages correlate to the augmenting severity of tau pathology [11]. This seminal work sprouted the so-called “Braak hypothesis” of transmission of tau pathology in AD. Since then, evidence from in vivo and in vitro studies has emerged, supporting the proposal, that tau extends throughout the brain via interconnected pathways in a mode similar to the spread of pathological prion proteins [6,12]. Additionally, tau spreading along axonal pathways has been demonstrated in human patients with tauopathies [13]. This hypothesis is also applicable to other proteins, including Aβ and alpha synuclein (α-Syn) [14,15,16,17]. The notion follows that pathological tau aggregates are released from one cell and internalized by a neighboring cell whereby they act as templates for misfolding of native tau protein in healthy cells in a process termed “template misfolding” [6].

Tau pathology was observed when brain extracts from mutant tau (P301S) transgenic mice or human AD brains were injected into wild-type tau-expressing mice at both the site of injection initially and interconnected brain regions at later time points [18,19,20,21]. Similar results were also described in transgenic mice expressing the human P301S tau mutation [22,23]. In other studies, synthetically generated pre-formed fibrils (PFF) of tau were injected in tau transgenic mouse models, which resulted in formation of tau pathology at the site of the injection along with propagating to connected brain regions in a time-dependent manner [24,25,26,27].

In vitro models also report that tau pathology propagates from donor cells to neurons mediated via trans-synaptic contacts [28] and increased neuronal activity [29,30]. Moreover, tau aggregates were taken up by cultured cells in a process that may be mediated by heparan sulfate proteoglycans and low-density lipoprotein (LDL) receptor-related protein 1 (LRP1) [31,32]; seeded aggregation of native intracellular tau and newly formed tau aggregates could transfer between co-cultures [33,34].

### 1.3. Spreading in Organotypic Brain Slices

Organotypic brain slice cultures are an excellent and valuable approach to study neurodegenerative diseases [35]. The system remarkably recapitulates structural morphology of the tissue and is amenable to manipulation via addition of substances to the media or live imaging of disease processes. Several groups, including ours, have successfully utilized slice cultures to investigate disease processes of AD [36,37,38,39,40,41].

Moreover, slice cultures have been employed to explore the prion-like propagation of Aβ and tau [42,43,44]. Recently, our group has explored the spreading of Aβ and α-Syn using organotypic coronal and sagittal slices, respectively, from wild-type and transgenic mice in an effort to model the transmission of these proteins [17,45]. Therefore, the aim of this study is to model the spreading of mutated P301S aggregated tau (P301S aggTau) using hippocampal coronal slices from wild-type mice to elucidate the general spreading characteristics and underlying mechanisms of spread between different cell types and regions. Collagen hydrogels are used to achieve a localized, stable and slow protein release. Additionally, we verify the effectiveness of collagen hydrogels as a protein delivery system for organotypic brain slices.

## 2. Materials and Methods

### 2.1. Animals

In this study, wild-type (C57BL/6N) mice were used. They were housed in the animal facility at the Medical University of Innsbruck with open access to food and water under 12/12 h light-dark cycles. Adults (1 male, 1 female) were housed per cage and the pups were kept in the same cage until post-natal day 8–10. The animals used for the organotypic brain slices were randomized between the groups, irrespective of sex. The Austrian Ministry of Science and Research (BMWF-66.011/0120-II/3b/2013 and 66.011/0055-WF/V/3b/2017) approved all animal experiments. The ethical principles of the 3 R’s rules were followed in our experiments and slice preparation is defined as organ removal as opposed to animal experiments. All animal-related lab work was in line with the Austrian and international guidelines on animal welfare and experimentation.

### 2.2. Tau Proteins

In this study, we used the following tau proteins: recombinant human tau (FL tau, R&D Systems, Minneapolis, MN, USA, SP-495), recombinant human tau Δ306-311 (R&D Systems, Minneapolis, MN, USA, SP-500) and recombinant human tau (mutated P301S) protein aggregate (active) (Abcam, Cambridge, UK, ab246003).

### 2.3. Organotypic Brain Slice Cultures

Organotypic brain slices were prepared as reported in detail in previous studies [45,46,47]. Briefly, postnatal day 8–10 C57BL/6N pups were rapidly decapitated, and the brains were removed under sterile conditions. The cerebellum was dissected, and the brains were glued (Glue Loctite 401) onto the chuck of a water-cooled vibratome (Leica Biosystems, Nussloch, Germany, VT1000A). Coronal slices (170 μM) were cut at the hippocampal level using a commercial razor blade in sterile preparation medium (16 mg/mL MEM/HEPES (Gibco, Thermo Fisher Scientific, Vienna, Austria, 11012-044), 0.43 mg/mL NaHCO_3_ (Merck-Millipore, Darmstadt, Germany, 144-55-8), pH 7.2) in the hippocampal region. Two slices were carefully placed on extra-membranes (Merck-Millipore, Darmstadt, Germany, HTTP02500) inside 0.4 μM membrane inserts (Merck-Millipore, Darmstadt, Germany, PICM03050) in a sterile 6-well plate (Greiner-Merck, Darmstadt, Germany, 657160). Each well contained 1 mL of sterile culture medium (16 mg/mL MEM/HEPES (Gibco, Thermo Fisher Scientific, Vienna, Austria, 11012-044), 0.43 mg/mL NaHCO_3_ (Merck-Millipore, Darmstadt, Germany, 144-55-8), 6.25 mg/mL glucose (Merck-Millipore, Darmstadt, Germany, 1083371000), 25% Hank’s solution (Gibco, Thermo Fisher Scientific, Vienna, Austria, 24020-091), 10% heat-inactivated horse serum (Gibco, Thermo Fisher Scientific, Vienna, Austria, 16050-122), 1% glutamine 200 mM stock solution (Merck-Millipore, Darmstadt, Germany, 1002890100), 1% of antibiotic-antimycotic (Gibco, Thermo Fisher Scientific, Vienna, Austria, 15240062 pH 7.2). The slices were incubated for a maximum of 9 weeks at 37 °C and 5% CO_2_. The medium was changed weekly until they were fixed in 4% paraformaldehyde (PFA) for 3 h and stored in PBS/0.1% Na-Azide at 4 °C.

### 2.4. Collagen Hydrogels and Loading

Collagen hydrogels were prepared as described previously [45,47,48]. The solution for collagen hydrogels was prepared by mixing 0.4 mg of type I bovine collagen solution (Sigma Aldrich-Merck, Darmstadt, Germany, 804592) and 0.3 mg of poly (ethylene glycol) succinimidyl succinate with phosphate-buffered saline (PBS) in a total volume of 200 μL. The same amount of PBS was added in place of the tau proteins to produce empty collagen hydrogels. Samples were kept on ice throughout to prevent premature polymerization of the hydrogels. The pH of the collagen hydrogel solution was set to 7.2 and 2 μL of the solution was pipetted on previously UV-sterilized Teflon-tape-coated glass slides. Subsequently, the collagen hydrogels were incubated for 15 min at 37 °C followed by immediate application onto organotypic brain slices. Collagen hydrogels were fluorescently visualized using a green fluorescent AlexaFluor-488 antibody (Invitrogen, Thermo Fisher Scientific, Vienna, Austria, A11029).

### 2.5. Hyperphosphorylation and Aggregation of Tau

To generate hyperphosphorylated tau, the different tau proteins (1 μg) were incubated with 2 μL of glycogensynthase-kinase-3β (GSK-3β) stock (Sigma Aldrich-Merck, Darmstadt, Germany, G4296) in 10.5 μL of tau kinase buffer (40 mM HEPES, 5 mM EGTA, 3 mM MgCl_2_, pH 7.6) including 2 mM ATP overnight at 37 °C. Hyperphosphorylated tau proteins were either loaded onto Western blots or used in collagen hydrogels [49]. Recombinant human tau was aggregated by incubating 600 ng of the protein in 10 μL of EDTA-free Protease Inhibitor Cocktail (PIC, Sigma Aldrich-Merck, Darmstadt, Germany, P-8340), 10 μL of 55 μM heparin solution with 74 μL of PBS in 100 μL of aggregation solution. The solution was incubated for 72 h at 37 °C. The efficiency of the aggregation was checked using Western blot analysis.

### 2.6. Western Blot

Western blots were carried out as described in prior publications [17,45]. Organotypic brain slices were scraped off from extra-membranes into Eppendorf tubes along with 80–120 μL of EDTA-free PIC (Sigma Aldrich-Merck, Darmstadt, Germany, P-8340). Slices were homogenized by sonication with an ultrasonic device (Hielscher Utrasonic Processor, Teltow, Germany) followed by centrifugation at 14,000× *g* for 10 min at 4°C and the supernatant was collected. The amount of total protein per sample was determined using the Bradford assay with Coomassie brilliant blue G-250 dye (Bio-Rad, Vienna, Austria, #1610406). Samples were loaded onto 10% Bis-Tris polyacrylamide gel (Invitrogen, Thermo Fisher Scientific, Vienna, Austria, NP0301BOX) in either a native or a denatured (10 min at 70 °C with 2 μL of sample-reducing agent (Invitrogen, Thermo Fisher Scientific, Vienna, Austria, NP000) state. Electrophoresis was carried out at 200 V for 35 min.

Samples were subsequently electrotransferred onto PVDF membranes (Merck, Darmstadt, Germany, ISEQ00010) for 20 min at 25 V in a semi-dry transfer cell (Thermo Fisher Scientific, Vienna, Austria). Blotting was conducted using a WesternBreeze Chemiluminescent immunodetection system (Invitrogen, Thermo Fisher Scientific, Vienna, Austria). Blots were blocked with blocking buffer for 30 min and incubated overnight at 4 °C with primary antibodies against the following: Tau5 (1:1000, Invitrogen, Thermo Fisher Scientific, Vienna, Austria, AHB0042), anti-tau phospho 396 (1:10,000, BioLegend, THP Medical Products, Vienna, Austria, 807401), tau monoclonal HT7 (1:500, Invitrogen, Thermo Fisher Scientific, Vienna, Austria, MN1000), neurofilament (1:10,000, Novus, THP Medical Products, Vienna, Austria, NB300-135), oligodendroglial marker MOG (1:2000, Proteintech, 12690-1-AP), microglial marker CD11b (1:2000, Proteintech, THP Medical Products, Vienna, Austria, 20991-1-AP), catalase (1:10,000, Thermo Fisher Scientific, Vienna, Austria, PA1-28372), astroglial marker GFAP (1:2000, Merck-Millipore, Darmstadt, Germany, AB5541) and actin (1:1000, Sigma Aldrich-Merck, Darmstadt, Germany, A2066) as a loading control. Thereafter, blots were washed and incubated with alkaline-phosphatase-conjugated secondary antibodies (anti-mouse for Tau5 and HT7, anti-chicken for GFAP, anti-rabbit for all others) for 30 min at room temperature. Following brief washing of the blots, they were incubated for 15 min in CDP-Star chemiluminescent substrate solution (Roche, Basel, Switzerland) and visualized with a cooled CCD camera (SearchLight, Thermo Fisher Scientific, Vienna, Austria).

### 2.7. Release Experiments and Tau Detection

Collagen hydrogels were prepared with either different tau proteins in a final concentration of 10 ng/μL or empty PBS load as described above. Three collagen hydrogels were placed on small rectangular pieces of Parafilm, which were then placed face down in 500 μL of sterile culture medium in 4-well plates (Nunc, Thermo Fisher Scientific, Vienna, Austria, 144444). Medium was collected in a time-dependent manner until 8 weeks. The total amount of tau protein released into the medium was measured using the automated Lumipulse Technology (Fujirebio, Ghent, Belgium, G600II).

### 2.8. Immunohistochemistry

Immunostainings were performed as detailed in prior studies [17,45]. To verify tau immunostainings, 5 µL of FL tau (500 ng) was slowly infused into the right cortex of a freshly dissected postnatal (P8-10) wild-type mouse brain via an injection syringe. The whole brain was then post-fixed in 4% PFA overnight. Cryosections were generated and analyzed by immunohistochemistry for tau (see Appendix A).

Fixed brain slices were washed 3 × in PBS and incubated in PBS-0.1% Triton (T-PBS) for 30 min at 20 °C on a shaker. Slices were then incubated with PBS-20% methanol-1% H_2_O_2_ to quench endogenous hydrogen peroxidase binding for 20 min at 20 °C when using biotinylated secondary antibodies. Following washing the slices 3× with PBS, they were blocked with T-PBS-0.2% bovine serum albumin (BSA, Serva, Heidelberg, Germany, 11930.03)-20% horse serum for 30 min at 20 °C. If the primary antibody was generated in mouse, an extra blocking step was performed by incubating the slices with PBS-mouse on mouse (M.O.M., Vector Laboratories, Newark, NJ, USA, MKB-2213) at a concentration of 1 drop per 2.5 mL for 1 h at 20 °C. Afterwards, slices were incubated with primary antibodies diluted in T-PBS-0.2% BSA for 2 days at 4 °C. The following antibodies were utilized to detect tau immunoreactivity: Tau5 (1:250, Invitrogen, Thermo Fisher Scientific, Vienna, Austria, AHB0042) and tau monoclonal HT7 (1:500, Invitrogen, Thermo Fisher Scientific, Vienna, Austria, MN1000). Neurons were labelled using an anti-neurofilament antibody (1:10,000, Novus, THP Medical Products, Vienna, Austria, NB300-135). Subsequently, the slices were washed with PBS and incubated with the appropriate conjugated biotinylated (diluted by 1:200) or fluorescent secondary antibodies (diluted by 1:400) in T-PBS- 0.2% BSA for 1 h at 20°C while shaking. When using a biotinylated secondary antibody, the slices were rinsed 3× with PBS and incubated with avidin-biotin complex solution (Elite ABC-HRP Kit, Vector Laboratories, Newark, NJ, USA, PK-6100) for 1 h at 20 °C. Finally, the slices were washed with 50 mM Tris-buffered saline (TBS) and then incubated in 0.5 mg/mL 3,3′-diaminobenzidine (DAB, Sigma Aldrich-Merck, Darmstadt, Germany, D7304)-TBS-0.003% H_2_O_2_ at 20 °C in the dark until a signal was detected. Upon the appearance of a DAB signal, the reaction was stopped by adding TBS. The slices were washed 3× in PBS and mounted with Mowiol onto glass slides.

When using a fluorescent secondary antibody, the slices were washed 3× with PBS prior to and following incubation with either Alexa-488 (green fluorescent, Invitrogen, Thermo Fisher Scientific, Vienna, Austria, A11029) or Alexa-546 (red fluorescent, Invitrogen, Thermo Fisher Scientific, Vienna, Austria, A11030/A11040) antibodies. All slices were counterstained with the blue fluorescent nuclear dye DAPI (1:10,000). After a final washing step with PBS, slices were also mounted with Mowiol onto glass slides. Immunostainings were visualized with a light microscope (Olympus BX61, Vienna, Austria) or inverse microscope (Leica DM IRB, Austria). Images were captured and analyzed with OpenLab software (Version 5.5.0, Improvision Ltd., Perkin Elmer, Vienna, Austria).

Confocal microscopy was conducted using an SP8 confocal microscope (Leica Microsystems, Wetzlar, Germany) with 1.3 NA glycerol objective. The emission of AlexaFluor-488 was detected from 493 to 556 nm and AlexaFluor-546 was detected from 566 to 628 nm. Subsequently, images and z-stacks were captured and deconvoluted by Huygens Professional software (Scientific Volume Imaging, Hilversum, Netherlands) and reconstructed in 3-D with Imaris software (Version 8.2, Oxford Instruments, Abingdon, UK).

### 2.9. Lactate Dehydrogenase (LDH) Assay

To assess the viability of brain slices during the incubation period, an LDH assay was performed. Media were collected from slices with either empty or P301S aggTau-loaded collagen hydrogels after 8 weeks and stored at −20 °C until analysis. The concentration of LDH released into the media was quantified in a 96-well format using the protocol described in the CheKine™ Lactate Dehydrogenase Assay Kit (Abbkine, THP Medical Products, Vienna, Austria, KTB1110). The absorbance values were measured at 450 nm with a microplate ELISA reader (Zenyth 3100 ELISA reader).

### 2.10. Data Analysis and Statistics

Images were acquired at 4× magnification from the light microscope using the same brightness and automatic exposure time settings from OpenLab image acquisition software. The images from slices with a field size of 2438 × 1829 µm were analyzed on a blinded basis using ImageJ (1.42q, National Institutes of Health, Bethesda, MD, USA) and the mean value of selected areas was taken as a measure of optical density. Data were extracted from three regions of interest: Area 1 (location of the collagen hydrogel), Area 2 (left and right cortex near the hippocampus) and Area 3 (ventral part of the slices). One image was analyzed from Area 1, two images from left and right hemispheres from Area 2 and 3 per slice. Final optical density was calculated by subtracting slice background from the initial values. Statistical analysis was performed by one-way ANOVA with a Fisher’s LSD post hoc test, where *p* < 0.05 represented significance. A student’s *t*-test with equal variance was used when comparing two groups. Data values are presented as mean ± standard deviation (SD), unless stated otherwise. Values in parentheses in bar graphs denote the number of analyzed animals in each experimental group.

## 3. Results

### 3.1. Culturing of Organotypic Brain Slices with Collagen Hydrogels

Coronal organotypic brain slices were cultured with a thickness of 170 µm at the hippocampal level, which flattens during the culturing period (Figure 1A). Collagen hydrogels (2 µL bolus) were placed above the hippocampus to load P301S aggTau (Figure 1A,B). They were approximately 2 mm in size and could be visualized after loading with a fluorescent Alexa-488 antibody (Figure 1C). Tau was fluorescently detectable in collagen hydrogels loaded with P301S aggTau through immunofluorescence detection with the Tau5 antibody (Figure 1D). Collagen hydrogels placed onto the brain slices were visible after 4 days but appeared fragmented and were completely degraded after 14 days in the culture releasing tau (Figure 1E,F).

### 3.2. Characterization of the Tau Proteins

Tau proteins were characterized by performing Western blots. In the present study, three tau proteins were initially tested: full-length (FL) tau, Δ306-311 tau and P301S aggTau. The Tau5 antibody detected as little as 20 ng FL tau per lane, which was seen as a 60 kDa protein (Figure 2A). FL tau could be successfully aggregated and appeared as a smear of proteins with a molecular weight of >60 kDa (Figure 2B). FL tau and Δ306-311 tau were clearly detectable as a 60 kDa protein (Appendix A). P301S aggTau appeared as a smear of large proteins from >50 kDa (Figure 2C). The enzyme GSK-3β successfully hyperphosphorylated the tau proteins, which were visualized by using the phospho-tau 396 antibody. Similar to the unphosphorylated tau proteins, FL tau and Δ306-311 tau were seen as a 60 kDa protein (Appendix A) and P301S aggTau as a smear with >60 kDa size (Figure 2D).

### 3.3. Loading of Tau Proteins into Collagen Hydrogels

Surprisingly, there was no protein detectable with the Tau5 antibody when FL tau, aggregated FL tau or Δ306-311 tau were loaded into collagen hydrogels (Appendix A). The addition of protease inhibitors in the collagen hydrogel solution did not lead to an improvement. However, loading of P301S aggTau into collagen hydrogels displayed a positive detection of tau at the expected 60 kDa mark in the presence and absence of protease inhibitors (Figure 2E). Collagen hydrogels loaded with hyperphosphorylated FL tau or Δ306-311 tau could not be detected by Western blot (Appendix A), but again hyperphosphorylated P301S aggTau loaded in collagen hydrogels was detectable (Figure 2E). When these P301S aggTau protein-loaded collagen hydrogels were placed onto brain slices for 4 or 8 weeks, a slightly weaker band was discernible at 60 kDa with the Tau5 antibody (Figure 2F). A second tau antibody (HT7) detected P301S aggTau (500 ng protein/lane), but it did not detect P301S aggTau loaded into collagen hydrogels. However, when P301S aggTau was loaded in collagen hydrogels and placed on brain slices, a clear band was visible at 60 kDa after 4 and 8 weeks incubation time. Interestingly, a smaller 30 kDa tau fragment was additionally detected after 8 weeks (Figure 2G). Based on these data, we chose to focus on P301S aggTau for further experiments with the brain slices.

### 3.4. Release of Tau from Collagen Hydrogels

In order to measure the protein release from collagen hydrogels, three FL tau or P301S aggTau-loaded collagen hydrogels per time point were placed on a small piece of Parafilm, which was then incubated upside down in sterile slice medium for 8 weeks. The medium was collected weekly. As a control, empty collagen hydrogels were also included. P301S aggTau was released from collagen hydrogels into the medium over a period of 8 weeks but never reached the maximum level of release (120 ng/mL) (Figure 3). In comparison, the release of FL tau was lower, and no tau was detected in the negative control.

### 3.5. Viability of Organotypic Brain Slices

Next, we wished to investigate if the application of collagen hydrogels on the slices affects the overall viability of the slices. This exploration was assessed using Western blotting to probe for cell markers. The blots showed a stable expression of neuronal neurofilament, oligodendroglial MOG, microglial CD11b and astroglial GFAP after the 9 weeks of the culturing period (Figure 4A). Catalase expression was weak and variable between samples (Figure 4A). Upon quantification of the band intensity from the Western blots, there was no significant difference between the slices loaded with empty or P301S aggTau-loaded collagen hydrogels (Figure 4B). Moreover, the amount of tau was measured in the samples utilizing the automated Lumipulse technology. There was significantly more tau in the samples from the slices loaded with P301S aggTau compared to empty hydrogels (*p* = 0.012) (Figure 4C). LDH release into the media correlates with the degree of cell death and thus, this assay reported the extent of cell survival. LDH release is comparable and not significantly different between slices with empty or P301S aggTau-loaded hydrogels after 8 weeks of culturing period (Figure 4D).

### 3.6. Immunostainings for Tau

To verify the tau immunostaining through the Tau5 antibody, a supplementary experiment was performed as an initial step. A freshly dissected postnatal mouse brain was injected with 5 µL of FL tau (500 ng), immediately post-fixed and then cryostat sections were produced. These sections were immunostained with the Tau5 antibody (Appendix A). Strong immunoreactivity (detected through DAB substrate and fluorescence) was observed in the injection side. Conversely, only a weak background on the contralateral side and in slices incubated without Tau5 antibody as a negative control was discernible.

Tau immunostaining was apparent inside collagen hydrogels in the slices that were immediately post-fixed at the 0-week time point (Figure 5A,F). By 1–2 weeks, the tau immunostaining disappeared inside the collagen hydrogels and instead appeared in the hippocampal areas (Figure 5B,C,G,H). After an incubation of 3–4 weeks, the immunostaining extended into the ventricles and cortex (Figure 5D,I), and after 8 weeks of incubation, prominent Tau5 immunostaining was observed in the piriform/entorhinal cortex (Figure 5E,J). This pattern of spreading was also similar, as detected with the HT7 antibody, which specifically recognizes the human tau isoform (Figure 5K–T).

Quantification of the Tau5 staining was performed in three different areas of interest by measuring optical density after 8 weeks of incubation (Table 1). Slices incubated with medium alone or with application of 2 µL PBS onto the slice revealed only a background of approximately 30 optical density units in all three areas (Table 1). These groups did not significantly differ from brain slices where an empty collagen hydrogel was placed (Table 1). Application of collagen hydrogels loaded with FL tau or Δ306-311 tau served as additional controls and the optical densities do not differ from those with the empty hydrogels (Table 1). Only the application of collagen hydrogels loaded with unphosphorylated and hyperphosphorylated P301S aggTau showed highly significant Tau5 immunoreactivity in the ventral part of the brain slices (Table 1), but not in the other two areas of interest.

### 3.7. Spread of P301S aggTau along Neuroanatomically Connected Pathways

In order to examine further the details of the transmission of tau pathology, slices with P301S aggTau-loaded collagen hydrogels were subject to a lesion using a cell scraper one week after the application of the hydrogels. The lesion was made in the middle of the slices, and was visible to the naked eye (Figure 6B). As a negative control, slices with P301S aggTau-loaded hydrogels were not subject to a lesion (Figure 6A). Subsequently, immunostaining for Tau5 followed the culturing period of 8 weeks and optical density data were extracted from Area 3. Interestingly, in slices with a lesion, tau immunostaining was significantly reduced in the ventral regions of the slices compared to the control slices, indicating that P301S aggTau spreads to the ventral parts of the slices through interconnected pathways (Figure 6C).

### 3.8. Localization of P301S aggTau

Lastly, in order to investigate the cellular localization of P301S aggTau in brain slices, HT7 immunostaining and subsequent confocal microscopy were performed. The spread of P301S aggTau was localized in neuronal cells with a positively stained cell body and neuronal afferent and efferent extensions (Figure 7A–D). No signal was observed from the red channel, indicating the specificity of tau staining and lack of a considerable background (Figure 7B). In 3-D reconstructions of deconvoluted confocal microscopy z-stacks, P301S aggTau was localized in the somatic compartment and in elongated structures resembling axons (Figure 7E). Apart from a single axon, multiple smaller tube-like structures were also stained for P301S aggTau (Figure 7F). Quantification of the mean length and width of the “microtubes” ± SEM yielded 24.1 ± 2.4 µm (10) and 1.3 ± 0.1 µm (10), respectively. To further ascertain the cellular identity of the P301SaggTau staining, double-immunostaining of slices with tau and neuronal marker neurofilament was performed, which exhibited a colocalization between the two proteins, highlighting that P301S aggTau is found in neuronal cells (Figure 7G–J).

## 4. Discussion

In the current study, the spreading characteristics of exogenously applied tau proteins on coronal organotypic brain slices produced from whole mouse brains were investigated. Collagen hydrogels were utilized as a protein delivery system to achieve a slow, controlled release of tau proteins onto the brain slices. We report that P301S aggTau, in both the unphosphorylated and hyperphosphorylated forms, significantly spreads to the ventral parts of the slice over a period of 8 weeks in culture. This spreading is interrupted in slices where the neuronal connections were disrupted by making a lesion.

### 4.1. Viability of Organotypic Brain Slices to Study Tau Spreading

Organotypic brain slices are increasingly used as an experimental approach to model neurodegenerative diseases including AD, Parkinson’s disease (PD), Huntington’s disease and amyotrophic lateral sclerosis (ALS) [44]. They represent a remarkable system to investigate various facets of neurodegeneration. Specifically, changes can be monitored after particular time points, such as after the addition of exogenous proteins of interest. A prime advantage of using slice cultures over other in vitro approaches such as primary cells or simple cell cultures is that the cytoarchitecture is largely intact with all cell types in the brain [35]. Hence, slices capture the physiological state in spite of loss of connectivity owing to axotomy. At the same time, slices can be produced from any mouse model, including transgenic models that are lethal beyond the perinatal period, which opens up the type of questions that can be investigated. Another advantage of slice cultures over in vivo models is that multiple slices can be produced from the same animal and there is no need to induce disease, which contributes to reducing the overall number of animals.

In order to study the spreading of P301S aggTau, it was crucial to examine the viability of organotypic slice cultures over the total incubation period of nine weeks. Collagen hydrogels were applied after one week of culture to allow for recovery time for the tissue to adhere and flatten to the membranes and they were maintained for eight weeks. This time point was chosen due to previous studies from our lab on Aβ and α-Syn proteins in collagen hydrogels on organotypic brain slices [17,45]. It is long enough to enable the spread of tau proteins but at the limits of overall slice viability, hence it makes for a reasonable compromise between the two requirements. To examine the viability of slices, a Western blot was performed to probe for various cellular markers with slices that were incubated with either empty or P301S aggTau-loaded collagen hydrogels for eight weeks. There was no significant difference in expression levels of neuronal marker neurofilament, oligodendroglial marker myelin-oligodendrocyte protein (MOG), microglial marker CD11b, and astroglial marker GFAP between these two groups. Astroglial marker GFAP displayed a fragmented pattern indicating the presence of several breakdown products, which has been previously reported in other studies and it might be a consequence of the physical stress to the tissue when generating the slices, which leads to activation of astrocytes at the borders of the slices [17,45]. The lack of a significant difference between the two groups also suggests that the application of P301S aggTau in the collagen hydrogels does not adversely affect the viability of the slices. Overall, the slices displayed good viability over the culturing time of nine weeks.

### 4.2. Collagen Hydrogels—Effectiveness as a Protein Delivery System

Slice culture experiments frequently apply exogenous compounds either directly on top of the slices or in the surrounding medium for a limited time period [38,39]. While both these methods are successful in studies, the spread of the compound is rather diffuse, and therefore, it is harder to control and localize it to one region of the slice. Coupled with the medium changes, it is difficult to regulate precisely the amount of the compound that is delivered to the slices. Other studies use microinjection techniques to deliver the compound to a specific target location. However, this requires specialized equipment and a certain slice thickness. Tau-containing exosomes were applied to hippocampal organotypic slice cultures in the media and it was reported that this method of delivery was more efficient than non-exosomal free tau [50]. This highlights that there is a need for an effective method to apply exogenous proteins to slices for neurodegeneration research. For these reasons, we chose to employ collagen hydrogels to deliver P301S aggTau in a local and specific manner. Collagen is the most abundant constituent of the extracellular matrix (ECM) and has a widespread use in many industries, particularly as a biomaterial in tissue engineering or reparative medicine [51,52]. In our lab, we have used collagen hydrogels due to their cost efficiency, versatility, biocompatibility and low immunogenicity to deliver Aβ and α-Syn proteins on organotypic brain slices [17,45].

Our results show, in line with previous studies, that collagen hydrogels loaded with the different tau proteins are subject to natural degradation over a period of 14 days, as evidenced by the disappearing fluorescent signal from the Alexa-488 antibody. Our release data demonstrate that both FL tau and P301S aggTau are released into the medium in a time-dependent manner. However, P301S aggTau clearly shows more release from the hydrogels compared to FL tau. This surprising observation could be owing to the differential interaction of the tau proteins inside the collagen and PEG matrix. Furthermore, the maximal P301S aggTau concentration was noted at six weeks, which proceeds to drop slowly for the seven and eight weeks time points. By these time points, the collagen hydrogels have degraded, and presumably, all the tau is released into the medium. It may be plausible that tau is degraded in the serum-containing media at the later time points in this slice-free experiment. Further experiments could shed more light on the levels of P301S aggTau released from collagen hydrogels at later time points. Taken together, the release data show that collagen hydrogels release tau proteins slowly into the medium over time.

### 4.3. Tau Proteins Used in This Study

This study initially examined three different tau proteins, which were characterized through Western blot: FL tau, Δ306-311 tau and P301S aggTau. The FL tau was chosen because it represents the complete version of the protein including four microtubule-binding domains and two N-terminal inserts [53]. Although all six isoforms are present in the adult human brain, only the FL tau isoform contains both hexapeptide motifs (VQIVYK and VQIINK) that are essential for tau aggregation and filament formation [54,55]. This explains why the FL tau isoform is more prone to aggregation than the other isoforms. Moreover, one of the essential hexapeptide motifs (VQIVYK) in Δ306-311 tau protein is deleted from the primary sequence, enabling one to examine if this tau protein would spread to the same extent as FL tau. Lastly, P301S aggTau was used in this study, as it is one of the mutations in human MAPT gene that leads to early onset frontotemporal dementia and manifests the neuropathological features of hyperphosphorylated tau [56,57].

Western blot analysis verified that FL tau, Δ306-311 tau and P301S aggTau were detectable by the Tau5 antibody at the expected 60 kDa size. Hyperphosphorylation of the tau proteins by GSK-3β was successful as evidenced by detection with the phospho-tau 396 antibody. When the tau proteins were loaded in collagen hydrogels and analyzed by Western blot, only P301S aggTau was detectable with the Tau5 and phospho-tau 396 antibody. This could be attributable to the interaction of the other tau proteins within the collagen and PEG matrix, possibly changing the conformation of the proteins in a manner that is hindering the recognition of these proteins in the Western blot. This result was also observed by another antibody (HT7), which detected P301S aggTau in slice extracts that were incubated with P301S aggTau-loaded hydrogels for four and eight weeks. The prominent appearance of a smaller breakdown product of tau at the 30 kDa size points to the processing or degradation of tau by the tissue over eight weeks. Since investigation into the differential interaction of tau proteins inside the collagen hydrogel matrix was beyond the scope of this study, we chose to focus on P301S aggTau and utilize the other two tau proteins as additional controls for further experiments.

### 4.4. Spreading of P301S aggTau to Ventral Areas

The time course experiment was performed to follow the pattern of P301S aggTau spread in hippocampal organotypic slices from the time of collagen hydrogel application to the end of the 8-week incubation period. Tau immunostaining was detected by Tau5 and HT7 antibody to ensure that the spread of exogenously applied tau was captured. Our results revealed that P301S aggTau is initially detected contained inside the collagen hydrogels at the 0-week time point. By eight weeks, considerable tau immunostaining was found in the ventral areas of the slices, potentially in the piriform/entorhinal cortex location. This spatio-temporal spread of P301S aggTau is in line with previous in vivo results whereby extracts from transgenic P301S mice and human AD brain were injected into wild-type mice and tau pathology was reported at the site of injection and neighboring brain areas [18,19,20,21]. Although the experimental setup is different between our experiments and in vivo data, our results highlight the spread of P301S aggTau in a time-dependent manner.

The optical density of tau-like immunoreactivity was quantified as a proxy measure of the intensity of tau immunostaining in three areas of interest: Area 1 (location of the collagen hydrogel), Area 2 (cortex region adjacent to the hippocampus) and Area 3 (ventral area of the slice). Slices with un- and hyperphosphorylated P301S aggTau-loaded collagen hydrogels displayed significantly higher optical density in Area 3 compared to slices with empty collagen hydrogels. These findings suggest that P301S aggTau has significant spreading potential. This is especially interesting considering that the aggregated tau could result in templated misfolding of the native, healthy tau along connected neuronal pathways in slices from wild-type mice.

Accumulating evidence reports that the prion-like propagation of tau pathology is dependent on functionally connected pathways [18,20,22,26,28]. Our results show that severing of hippocampal brain slices in the middle was detrimental to the spread of P301S aggTau to the ventral regions of the slices, as evidenced by the significantly decreased tau immunoreactivity in Area 3 (*p* < 0.001). This lends credence to the notion that pathological spread of tau is dependent on synaptically connected anatomical pathways. However, it has to be noted that the method of generating brain slices itself involves axotomy, which could potentially be a confounding factor in the results. Furthermore, the role of glial cells in the transmission of tau pathology should not be neglected since astrocytes internalize, gather tau prior to the appearance of clinical symptoms and can transfer tau oligomers to neighboring neuronal networks [58,59]. Microglia can also play a role in aiding tau propagation as their depletion considerably reduced pathological tau spreading in P301S tau transgenic mice [60]. Thus, tau pathology can extend via neuronal and non-neuronal routes.

### 4.5. Mechanism of Tau Spreading

Upon closer inspection with confocal microscopy, the results reveal that P301S aggTau is found in neurons in the ventral areas of the slices with a clear localization in the axons, dendritic arbor and cytosolic compartments. This was detected by HT7 antibody, and thus indicates that the detection is specific to the exogenously applied human tau. Three-dimensional rendering of z-stacks from confocal microscopy also highlights that P301S aggTau is taken up by neurons. However, further colocalization experiments with neuron-specific markers are required to completely ascertain the identity of the neuronal subpopulations.

Although tau is a microtubule-associated cytosolic protein, it has been found outside the cells in physiological conditions. Tau is present in the cerebrospinal fluid (CSF) of AD, particularly the phosphorylated form that appears to correspond to increasing disease severity [61,62]. Current evidence suggests that tau is actively secreted from neurons and this may have a yet unknown physiological role; these pathways can also be explored under pathological conditions. There are the four primary mechanisms of tau secretion and spreading: (1) direct translocation across the plasma membrane mediated by heparan sulfate proteoglycans (HSPGs) [63,64], (2) vesicular transport through exosomes and ectosomes [50,60,65], (3) release through the endolysosome and autophagy pathways [66], (4) direct cell-to-cell transfer via tunneling nanotubes [67,68]. From our confocal microscopy z-stacks, what appears to be tunneling microtubes positively stained for P301S aggTau could be observed. Nonetheless, further validation of this hypothesis is necessary by colocalization experiments with markers of tunneling microtubes such as actin and myosin 10.

### 4.6. Limitations of the Study

This study certainly has limitations. (1) In this study, we used slices from wild-type postnatal mice due to the low viability of slices obtained from adult mice and rats. This is incongruous with the fact that age is the greatest risk factor in the development of neurodegenerative diseases. Other groups have been successful in culturing hippocampal slices from old mice or 3× transgenic AD mice through the use of serum-free media [69,70]. Thus, it would be interesting to analyze the spread of tau proteins in slices from aged animals or transgenic AD animals, especially P301S mice. (2) Sporadic forms of AD are the most frequent type and the diagnosis is preceded by a preclinical stage that typically lasts for 20–30 years. Familial forms of AD constitute less than 1% of the cases [71]. Considering the immense time it takes for neuropathological features of Aβ plaques and NFTs to manifest, the incubation time of eight weeks in our experiments is evidently not comparable to the stages of human AD. Nonetheless, organotypic brain slices are a vital compromise to study the spreading of tau proteins in an open and amenable system. (3) Our study investigated the spread of P301S aggTau with the application of collagen hydrogels above the hippocampal region. According to the Braak staging hypothesis, tau pathology appears in the entorhinal cortex in the earlier stages and subsequently spreads to the hippocampal areas. Hence, it would be intriguing to analyze the spreading characteristics of P301S aggTau with the placement of collagen hydrogels in different locations, particularly the entorhinal cortex. (4) Lastly, owing to the yet unidentified reasons as to why FL tau and Δ306-311 tau were not detectable when loaded in collagen hydrogels, the question regarding the spread of these proteins in hippocampal organotypic slice cultures remains open-ended.

## 5. Conclusions

In conclusion, this study reports that collagen hydrogels are an effective experimental approach to study the spreading of tau proteins using organotypic brain slices. P301S aggTau possesses spreading potential as it spreads from the collagen hydrogels to the ventral areas of the slices in a time-dependent manner. This is supported by the significantly enhanced tau immunostaining in this region compared to the location of the hydrogels or the cortical areas. P301S aggTau appears to localize in neurons in the ventral regions. Furthermore, the transmission of P301S aggTau is dependent on neuroanatomically connected pathways. However, future experiments will elucidate the finer underpinnings of the prion-like propagation of tau proteins in tauopathies.

## Figures and Tables

**Figure 1 biomolecules-12-01164-f001:**
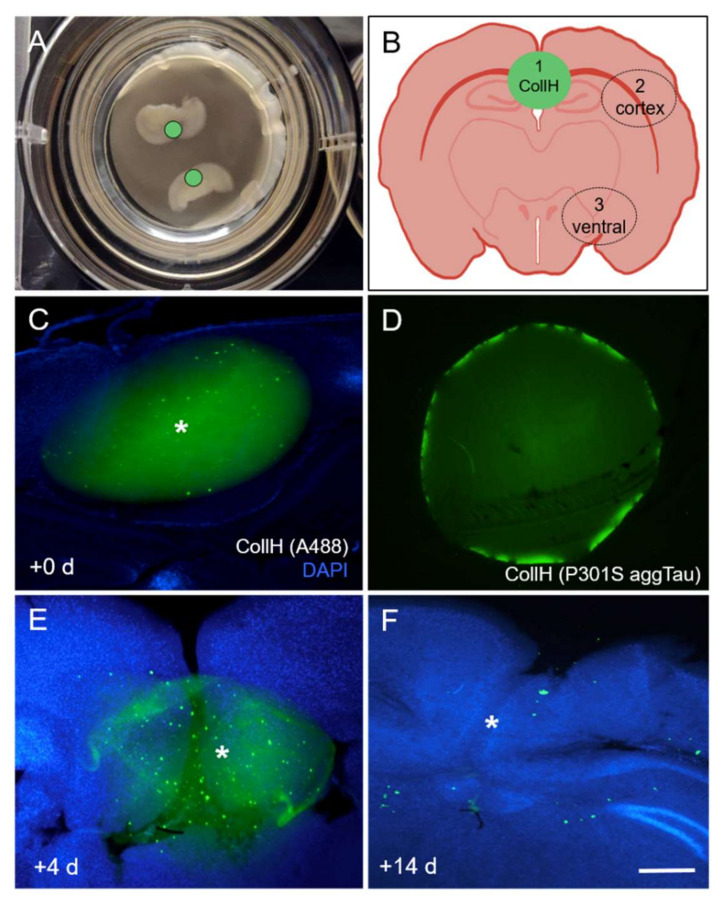
Characterization of collagen hydrogels. (**A**) A representative image showing coronal hippocampal organotypic brain slices (170 µm thick) on top of extra-membranes inside cell culture inserts. (**B**) A schematic demonstrating the placement of collagen hydrogels on the top of the slice and above the hippocampus region (green circle). The numbers indicate the three areas that were delineated to extract optical density after the incubation of brain slices with P301S aggTau-loaded collagen hydrogels. Area 1 is the location of the collagen hydrogel; Area 2 is the cortex and Area 3 is the ventral region of the slice. (**C**) Collagen hydrogels are approximately 2 mm in size and could be visualized after loading with a green fluorescent AlexaFluor-488 antibody. (**D**) P301S aggTau was detectable with the Tau5 antibody when it was loaded into collagen hydrogels. (**E**) Collagen hydrogels are visible after 4 days on organotypic brain slices in a slightly fragmented form. (**F**) No green fluorescent AlexaFluor-488 signal was observed after the collagen hydrogels were in culture 14 days, suggesting a complete degradation and release of tau proteins. The * denotes the location of the collagen hydrogel. Scale bar in F = 8890 µm in (**A**); 230 µm in (**C**–**F**).

**Figure 2 biomolecules-12-01164-f002:**
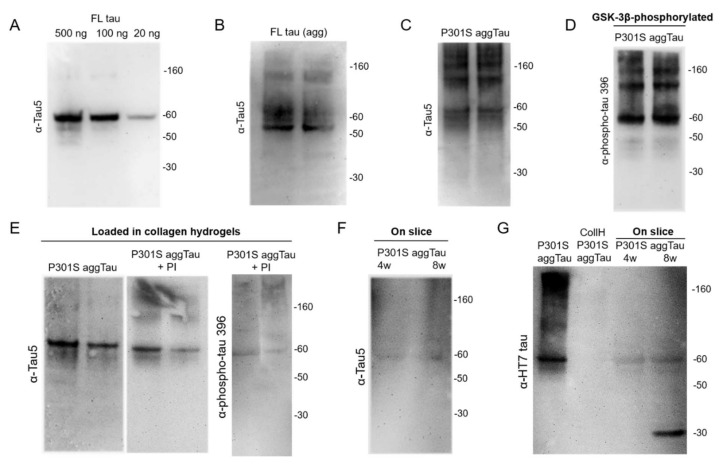
Characterization of the P301S aggTau using Western blots. (**A**) Immunostaining with the Tau5 antibody detects full-length (FL) tau in the range of 20–500 ng and the bands were observed at the expected size of 60 kDa. (**B**) FL tau aggregated (agg) was evident by the smear of proteins with a molecular weight >60 kDa. (**C**) P301S aggregated tau (P301S aggTau) appeared as a smear of large proteins >50 kDa. (**D**) GSK-3β hyperphosphorylated the tau proteins, which were detectable with the phospho-tau-396 antibody. P301S aggTau was detected as a smear of proteins in 50-160 kDa range. (**E**) P301S aggTau loaded in collagen hydrogels displays positive detection of tau at the expected 60 kDa mark, which was also observed when protease inhibitors were added. Hyperphosphorylated P301S aggTau is detected at the 60 kDa mark with the phospho-tau-396 antibody. (**F**) When P301S aggTau protein was loaded into collagen hydrogels and placed onto brain slices for 4 or 8 weeks, a slightly weaker band was visible at 60 kDa. (**G**) The tau monoclonal HT7 antibody detected P301 aggTau (500 ng/lane), but it did not detect P301S aggTau loaded into collagen hydrogels. However, when the P301 aggTau-loaded collagen hydrogels were applied on brain slices for 4 and 8 weeks, clear but weaker bands were seen at the 60 kDa mark. Additionally, a smaller tau fragment was observed at the 30 kDa size.

**Figure 3 biomolecules-12-01164-f003:**
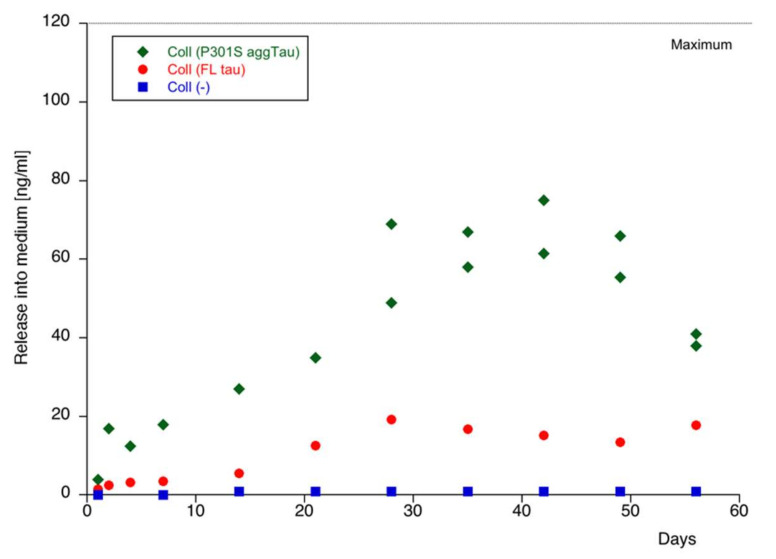
Release of tau proteins from collagen hydrogels. Full-length (FL) tau (red circles) or P301S aggTau (green diamonds) were loaded into collagen hydrogels and placed on small pieces of Parafilm face-down in culture media from 0 to 8 weeks. Empty collagen hydrogels (blue squares) were also incubated as a negative control. The media were collected weekly, and the level of tau was determined by a tau-specific ELISA. P301S aggTau was released more from the collagen hydrogels than the FL tau but it does not reach maximal release. No tau was detected from the empty hydrogels. The Lumipulse measurements were repeated from the same samples for Coll (P301S aggTau) for 4, 5-, 6-, 7- and 8-week time points. Values are reported as ng/mL.

**Figure 4 biomolecules-12-01164-f004:**
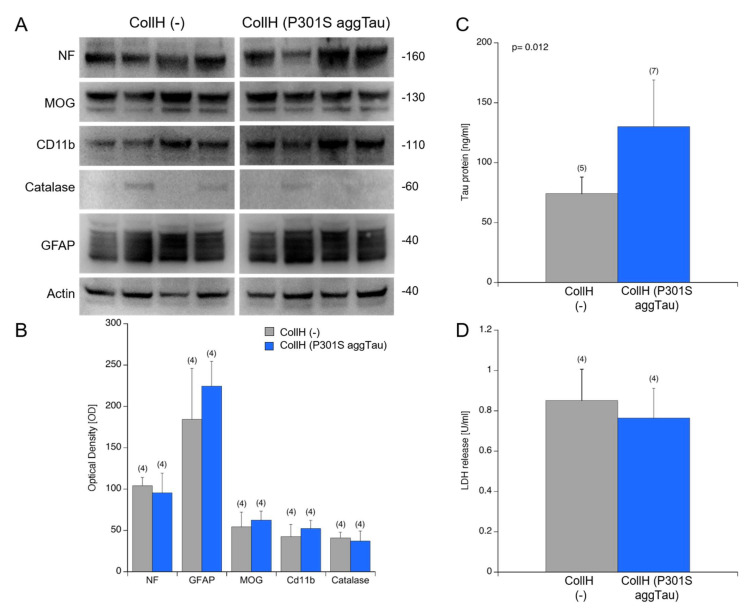
Effects of P301S aggTau on slice viability. (**A**) Slices were incubated with collagen hydrogels loaded with no load [CollH (−)] or P301S aggTau [CollH (P301S aggTau)] for 8 weeks. Subsequently, the slices were pooled together and probed for various cellular markers through a Western blot. Actin served as the loading control. There is no qualitative difference in the expression levels of neuronal NF, oligodendroglial MOG, microglial CD11b and astroglial GFAP between the CollH (−) and CollH (P301S aggTau) groups. Catalase expression is weak and variable in both groups. (**B**) Quantification of the band intensity reported no significant difference between CollH (−) and CollH (P301S aggTau) groups for any cellular marker. (**C**) Levels of tau protein were quantified through Lumipulse technology in slices from the CollH (−) and CollH (P301S aggTau) groups, which report a significantly higher tau level in slices incubated with P301S aggTau−loaded hydrogels (*p* = 0.012). (**D**) LDH release into the culture media was quantified and there was no significant difference between the slices from the CollH (−) and CollH (P301S aggTau) groups. Values are reported as mean ± SD and the number in parentheses represents the number of animals analyzed per group. A student’s *t*-test with equal variance was employed to compare mean values from both groups. Each lane in the Western blots represents an individual mouse sample. Values in parentheses indicate the number of analyzed animals.

**Figure 5 biomolecules-12-01164-f005:**
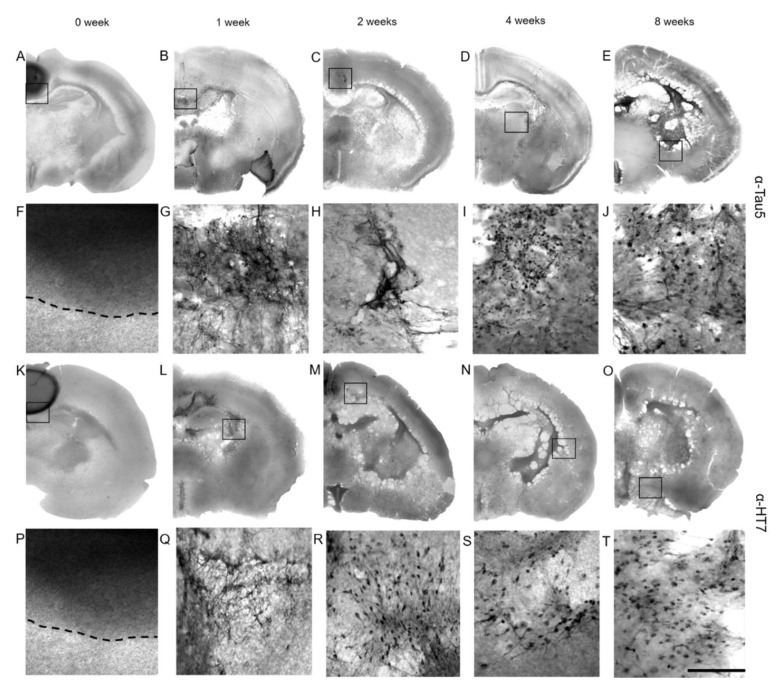
Spreading of P301S aggTau in organotypic brain slices. (**A**–**E**) Composite images of brain slices displaying the spread of P301S aggTau as detected by Tau5 antibody starting from inside of the collagen hydrogels at 0 week to the ventral parts of the slices at 8 weeks. (**F**–**J**) Magnified representative images of tau immunostainings of the black boxes indicated in (**A**–**E**). (**F**) A clear boundary of the P301S-loaded hydrogel can be seen from slices that were immediately post-fixed and immunostained after the application of collagen hydrogels. (**G**,**H**) Tau-specific positive immunostaining is observed in cells along with what appears to be neuronal connections after 1 and 2 weeks in culture. (**I**,**J**) After 4 and 8 weeks in culture, tau immunostaining appears in the ventral regions of the slices. (**K**–**O**) Composite images of brain slices of the spread of P301S aggTau as detected by the HT7 antibody, which is specific for the human tau isoform. The pattern of spread is similar to the one observed with Tau5 antibody. (**P**–**T**) Magnified representative images of the black boxes from the (**K**–**O**) images showing tau immunoreactivity spreading from the collagen hydrogels to the ventral areas. Scale bar in T = 930 µm in (**A**–**E**,**K**–**O**); 100 µm in (**F**–**J**,**P**–**T**).

**Figure 6 biomolecules-12-01164-f006:**
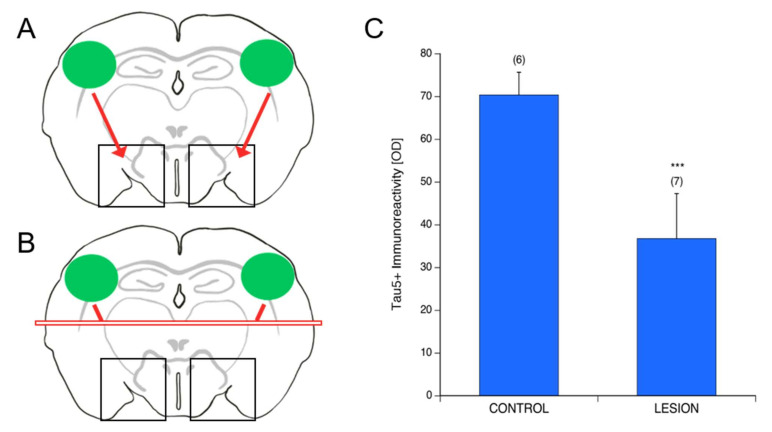
Spread of P301S aggTau along neuroanatomically connected pathways. (**A**,**B**) Schematic representations of the slices that were incubated with P301S aggTau-loaded collagen hydrogels, one each on the left and right hemisphere regions (green ovals). (**A**) Slices without a lesion and (**B**) slices that were subject to a lesion created by a cell scraper one week after the application of the collagen hydrogels (red rectangle). Subsequently, spreading of P301S aggTau to the ventral areas of the slices (Area 3) was analyzed (black boxes). (**C**) Quantification of the extracted optical density, which serves as a proxy for Tau5 positive immunostaining in Area 3. Slices with a lesion demonstrate significant decreased spreading of P301S aggTau to Area 3 compared to the control slices. The two groups were compared using a student’s *t*-test with equal variance (*** *p* < 0.001). Values in parentheses represent the number of animals analyzed. Data are presented as mean ± SD. Values in parentheses indicate the number of analyzed animals. Please note that we consider an optical density (OD) value of 21 to be unspecific background.

**Figure 7 biomolecules-12-01164-f007:**
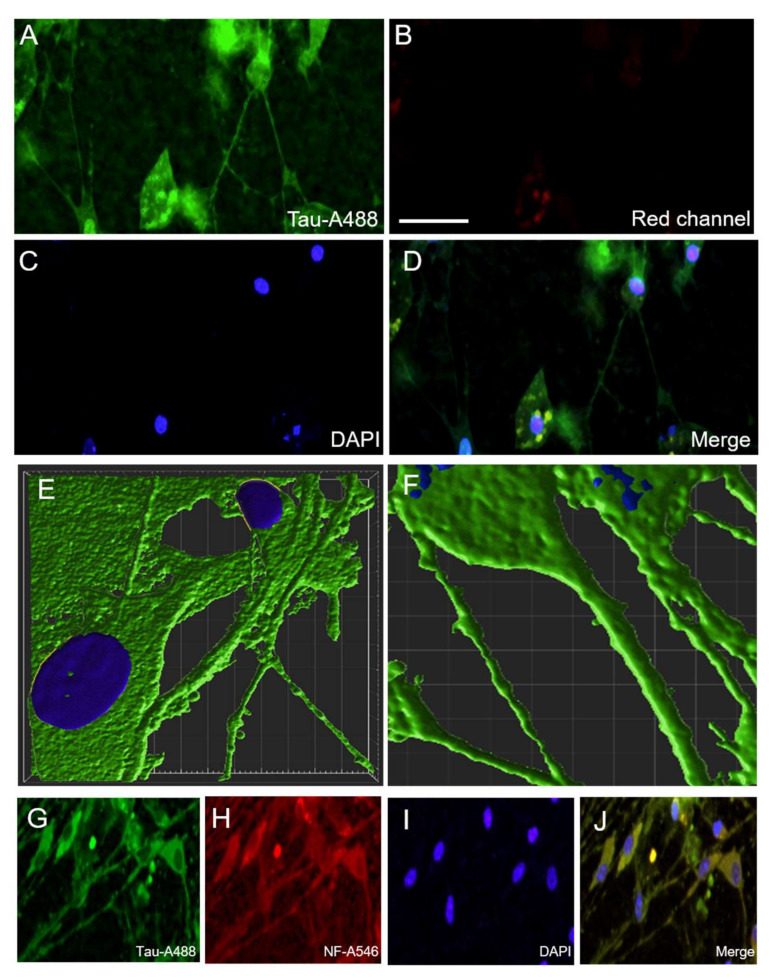
Localization of P301S aggTau in organotypic brain slices. (**A**) Representative image of P301S aggTau-specific positive fluorescent staining appeared in the ventral parts of the slices (green; AlexaFluor-488). (**B**) Little background staining was visible in the red channel. (**C**) Slices were counterstained for the nuclear dye, DAPI (blue fluorescence). (**D**) A merged image was generated, which reveals tau immunostaining in the cell bodies and connections between these cells. (**E**,**F**) Representative images of the 3-D reconstruction of confocal microscopy z-stacks stained for P301S aggTau (green) and the nuclei are shown in blue. (**G**–**J**) Slices were stained for tau (green; AlexaFluor-488), neurofilament as a neuronal marker (red; AlexaFluor-546) and nuclei (blue; DAPI). The merged image indicates that P301S aggTau colocalizes with neurofilament. Scale bar in B = 40 µm in (**A**–**D**); 6 µm in (**E**,**F**); 60 µm in (**G**–**J**).

**Table 1 biomolecules-12-01164-t001:** Quantification of tau spreading in organotypic brain slices.

	Area 1	Area 2	Area 3
*w*/*o* (−)	36 ± 8 (4)	23 ± 10 (4)	23 ± 8 (4)
PBS	26 ± 7 (4)	24 ± 7 (4)	21 ± 6 (4)
CollH (−)	28 ± 5 (12) vs.	24 ± 7 (12) vs.	29 ± 9 (12) vs.
CollH (FL tau)	43 ± 16 (5)	32 ± 17 (5)	33 ± 14 (5)
CollH (Δ306-311 Tau)	37 ± 20 (5)	29 ± 12 (5)	34 ± 18 (5)
CollH (P301S aggTau)	30 ± 21 (14)	30 ± 11 (14)	69 ± 19 (14) ***
CollH (P301S aggTau-P*)	25 ± 9 (3)	24 ± 10 (3)	72 ± 7 (9) ***

Organotypic slices at the hippocampal level were prepared. After 1 week in culture, collagen hydrogels (CollH) loaded with P301S aggTau and phosphorylated (P*) P301S aggTau were applied and further cultured for 8 weeks. As additional controls, slices were incubated without (*w*/*o*) treatment, phosphate-buffered saline (PBS, 2 µL), empty CollH (−), FL tau and Δ306-311 Tau. Slices were fixed and immunostained with the Tau5 antibody. Images were taken at 4× magnification with a field size of 2438 × 1829 µm and then were quantified using computer-assisted imaging. Tau-like immunoreactivity was evaluated in Area 1 (location of the CollH), Area 2 (left and right hemisphere cortex) and Area 3 (left and right ventral areas) (see also Figure 3B). Raw optical density measurements were inverted by correcting for slice background such that 0 represents white and 255 represents black. Values are generated from an average of two slices per animal and values in parentheses indicate the number of analyzed animals. Values are reported as mean ± SD of optical value. Please note that we consider an optical density value of 21 to be unspecific background. Statistical analysis was performed using a one-way ANOVA with a Fisher’s LSD post hoc test, where *p* values < 0.05 represent significance versus the CollH (−) group (*** *p* < 0.001).

## Data Availability

The data that support the findings of this study are available on request from the corresponding author.

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
