# Peer review of "Spreading of P301S Aggregated Tau Investigated in Organotypic Mouse Brain Slice Cultures"

_biomolecules, 2022, doi:10.3390/biom12091164_

Round 1

Reviewer 1 Report

Serious concerns:

References 1-9 do not make any sense. Therefore, I can’t tell if it the statements linked to the references are correct.

In general, I find that the papers referenced are not appropriate. 1)There is an overuse of reviews/reports and a lack of primary paper citations. For example, a report is cited for the spread of tau aggregates through connected brain regions. There are much better primary papers to support this statement from Adam Boxer, William Seeley, etc. 2) References don’t support statements. For example, on line 71-72 synuclein is discussed but not matched to any references cited (12-15). In another example, the statement on lines 80-83 discusses tau spreading but the references are for amyloid beta and synuclein spreading.

Minor concerns:

Line 55- “These structural features make tau particularly amenable to various post-translational modifications, such as phosphorylation, acetylation, ubiquitination and proteolytic cleavage.” I think you mean to say that the intrinsically disordered nature of tau makes it particularly amenable to PTMs?

Line 57 – “Phosphorylation is the best-characterised post-translational modification and hyperphosphorylation of tau leads to a pathological state and reduces the binding ability to microtubules and hence affecting the overall stability of microtubules [8]”. I disagree with the causal inference in this sentence. Hyperphosphorylation is linked to pathological states but it is also found in physiological conditions e.g. hibernation of animals. There are also work from Peter Baas’ lab that challenge the classical role of tau in regulating “overall stability” of microtubules. Perhaps it would be easiest to propose this statement as a hypothesis.

Please note that I stopped reviewing the manuscript at line 83 due to the serious concerns listed above.

Author Response

Please see response in PDF file.

Reviewer 2 Report

The authors report on brain slice cultures and the delivery of multiple Tau species to mimic tauopathy in a slice. 

Overall, the outcome is interesting and the data show the the method may be useful and it adheres to some elements of spreading previously reported for similar experiments using different proteins.

Some issues need attention:

1) the authors need to be careful with their interpretations. i) 301S is a model of FTD and not AD. so the interpretation of the data in the context of AD is overstated and needs revision. This is particularly clear in the last sentence of the Abstract. What novel insight has been gained by this work? It is purely a methodologically new approach but it does not provide any new mechanistic insight. 

2) The citation list is corrupted (possibly while formatting) and needs checking.

3) Statistically, I do not like the presentation of SEM. A more informative way is presentation of SD. Also, I am not quite clear what was taken as n in the analysis. Was it the number of animals used or was it the number of slices per treatment? If number of slices, were they from different animals? Also, what do numbers above bar chart denote?

4) Table 1 and are selection: The authors want to make a case for connectivity based transport of the seeded tau proteins. However, they do not show that spreading is lower into non-connected regions (areas 2and 3 are all anatomically interwoven with the release site). Also, the data in table one could be scewed and only the 301S mutation is producing positive results due to its heightened release from the hydrogel. Can this be accounted for? The labelling with the Tau5 pan tau antibody is not helpful here as it also stains for mouse tau so one cannot discriminate between human and mause tau relationship in the selected brain regions. Finally, the 50% tau labelling (Fig. 6C0 seen in controls is surprising. Is this because of the Tau5 AB/ Is this because the dissection/lesion was introduced after hydrogel administration? Is this becuase of tau release into the medium and free transport all over the slice? Or is it a mix of all of the above?

5) Viability: indeed, it seems that in terms of viability there is no difference although I noticed that there were no mitochondrial markers used or receptors and transmitter expression was not analysed. A more functional assay would have been instructive defining for example transmitter release or channel activation etc.  In gerenat, what is the defining criterion for good viability? There was no comparison between 1, 4 and 9 week old slices so deterioration cannot be determined!

6) Discussion: The discussion is far too long and it repeats too much of the results. I should be reduced by 50%. For example,  section 4.4 could be omitted as it does not contribute to the interpretation. In section 4.5, the authors repeat their case for spreading, especially for 301S. However, this needs to be moderated as no other species was tested  do to release issues and do to a missing non-connected area on the slice (see above).  

Author Response

Please see response in PDF file.

Reviewer 3 Report

This manuscript tested the P301S aggregated tau spreading in the brain through the connected regions by utilizing the organotypic mouse brain slice cultures. Collagen hydrogels loaded with P301S aggregated tau were applied on the brain slice, after 8 weeks culture, the immunostaining pictures at different times (0, 1, 2, 4, and 8 weeks) were captured to test the tau protein spreading to the slice ventral parts as a time-dependent manner. This manuscript shows the collagen hydrogel loading with tau protein applied on the organotypic mouse brain slice could be a possible tool to investigate the prion-like tau propagation in Alzheimers disease.

While an exciting and powerful tool to investigate the tau propagation in AD, and possibly other interesting molecules spreading in the brain, this manuscript could benefit from some modifications.

1.     The details of the experiment in the Materials and Methods” need to be clarified, such as how many mice were used in each group, and how many brain slices were used from each mouse (especially for each figure).

2.     Figure 3, how many experiments have been done? There is no error bar. For group Coll(-) and group Coll (FL tau), both are single data points at each time spot. But for group Coll(P301S aggTau), some time spots are single data points, but the others have two data points (4, 5, 6, 7, and 8 weeks).

3.     Figure 4A, does each lane in western blot represent one mouse sample?

4.     Figure 4A and 4D: why in 4A, Coll(-) and Coll (P301S aggTau) are 5 and 7 data points, but in 4D, Coll(-) and Coll(P301S aggTau) are both 4 data points?

5.     line 305: P301S or P301?

6.     For table 1, this paper is to show the tau spreading from area 1 to area 2, finally into area 3. It will be nice to have a table to show week 3 or 4 that P301S aggTau is with high density in area 2 but not entering into area 3 yet.

7.     For table 1, could you discuss more about why for group CollH(FL tau) and CollH(Δ306-311 Tau) could not observe high density at Area 3? Is that because they could not be transferred or just could not be detected with the antibody but do transfer? As discussion about the possible tau conformation change after the interaction within the collagen and PEG, does this mean this model could not be used for wildtype full-length tau?

8.     Line 710-720: References #1 to #8, all missing authors and titles.

Author Response

Please see response in PDF file.

Round 2

Reviewer 2 Report

This revision has taken account of all my previous comments.